# Shape, Resonant Frequency and Thermoelastic Dissipation Analysis of Free-Formed Microhemispherical Shells Based on Forming Process Modeling

**DOI:** 10.3390/mi13060913

**Published:** 2022-06-09

**Authors:** Yang Gao, Jiachao Zhang, Zhihu Ruan, Lin Meng, Jia Jia

**Affiliations:** 1Artificial Intelligence Institute of Industrial Technology, Nanjing Institute of Technology, Nanjing 211167, China; zhangjc07@njit.edu.cn (J.Z.); menglin@njit.edu.cn (L.M.); 2Jiangsu Province Engineering Research Center of IntelliSense Technology and System, Nanjing 211167, China; 3School of Instrument Science and Engineering, Southeast University, Nanjing 210096, China; 230179760@seu.edu.cn; 4School of Electronic Information, Jiangsu University of Science and Technology, Zhenjiang 212100, China

**Keywords:** microhemispherical shell, thermoelastic dissipation, process model, shell shape, free-form blowing process

## Abstract

Free-form microhemispherical shell resonators have the advantages of high quality factor and mass production. The shape of microhemispherical shells created via this process is based on a single mold and is difficult to adjust, which affects the resonant frequency and quality factor. In this paper, a process analysis model is established through in-depth analysis of the process mechanism and flow of the free-forming method. Based on this model, the influence of the designed preforming parameters on the shape, resonant frequency and thermoelastic dissipation of the microhemispherical shell are analyzed in detail, providing theoretical guidance for parameter design. The results show that the depth and the ratio of internal to external pressure of the substrate’s annular groove affect the height and thickness of the microhemispherical shell, and the structural thickness affects the thickness of the microhemispherical shell; these in turn affect the resonant frequency and thermoelastic dissipation of the microhemispherical shell resonator. In addition, the inner diameter of the substrate’s annular groove mainly affects the radius of the support column of the microhemispherical shell, and the influence on the resonant frequency and thermoelastic dissipation of the resonator is relatively low.

## 1. Introduction

Traditional macro-scale shell resonators have been successfully used to produce navigation-grade gyroscopes [1]. With the combined characteristics of high precision, long life and high reliability of traditional hemispherical gyroscopes, and the small size, light weight, mass production and easy realization of digitization and intelligence characteristics of MEMS devices, microhemispherical resonant gyroscopes have the potential for both small size and high precision. In addition, microhemispherical resonant gyroscopes can directly output the angle when working in full-angle mode, so they have the significant advantage of no angular velocity integration error and are currently recognized as one of the important development directions of high-performance miniaturized gyroscopes [2,3,4]. In addition, compared with macro hemispherical shells, the radius of a micro hemispherical shell is very small, and the structural deformation caused by centrifugal force during high-speed rotation can be ignored [5].

Microhemispherical shell resonators manufacturing can be divided into two types: thin film deposition and surface tension (including molding and free-forming) [6,7,8,9,10,11]. Between them, free-formed surface tension resonators not only has high quality factor due to the surface tension process, but also can be mass produced. However, the microhemispherical shell resonator is formed once under this process, and it is difficult to fine-tune later. Therefore, it is necessary to explore parameter design before forming the shape of the microhemispherical shell so as to more accurately design the parameters for the desired shape. At present, reports on this topic are relatively rare.

Changes to the microhemispherical shell shape cause changes to many properties of the resonator, including resonant frequency, energy dissipation and so on. The higher the resonant frequency, the stronger the environmental adaptability of the device, such as shock and vibration resistance. However, under the same electrostatic force, the higher the resonant frequency, the smaller the resonant amplitude of the resonator, which increases the difficulty of electrostatic driving and detection. Therefore, it is necessary to study the influence of preformed design parameters on the resonant frequency of microhemispherical shells in order to design parameters around the desired resonant frequency. In addition, high quality factor of microhemispherical shell resonators is a necessary prerequisite for the high accuracy and resolution of microhemispherical resonant gyroscopes. For microhemispherical shell resonators in four-antinode working mode, high quality materials, low resonant frequency (<100 kHz), vacuum environment and annealing can reduce the influence of support dissipation, internal dissipation, phonon interactions, fluidic damping and surface dissipation. Therefore, the major enegy dissipation mechanism in microshell resonators is thermoelastic dissipation (TED) [12].

The thermoelastic dissipation of resonators is one of the hot topics of research. Zener first described TED for simple beams in 1937. Nayfeh analyzed and simulated the quality factors due to thermoelastic damping ($Q_{TED}$) of microplates of general shapes and boundary conditions, and the results showed that the geometric properties of microplates can have a significant effect on $Q_{TED}$ [13]. Ref. [14] analyzed TED for complex geometries of micromechanical resonators by applying a fully coupled finite element simulation. Ref. [15] analyzed and calculated the TED in micromechanical resonators by fully coupled thermomechanical equations, finding the eigenvalues and eigenvectors of the uncoupled thermal and mechanical dynamics equations by using Comsol Multiphysics. The correctness of the theoretical analysis and simulation results were verified experimentally. Ref. [16] constructed complete quantitative and predictive models with finite element methods for the intrinsic energy dissipation mechanisms in MEMS resonators using full anisotropic representation of crystalline silicon. The results showed that TED is a more significant source of damping. Ref. [17] investigated the influence of design changes to the ring and support legs on thermoelastic damping by using a computational method based on COMSOL Multiphysics. Ref. [18] reported a numerical investigation of TED in micro-resonators with surface roughness by using COMSOL Multiphysics. It can be found that since thermoelastic dissipation is difficult to measure practically, the use of finite element simulation for TED research is a widely used and accurate method.

There are also some reports on the thermoelastic loss of microhemispheric shell resonators during the blowing process, mainly for molds. In 2014, the University of Michigan reported on the influence of resonator material, shell thickness, metallization-coating thickness and other factors on the thermoelastic dissipation of resonators [19]; in 2017, the effects of metallization-coating material parameters on thermoelastic dissipation in resonators were further reported [12]. Since there are few reports on the free-form blowing process, there are also few reports on the resonant frequency and energy dissipation of microhemispherical shells based on this process.

In this paper, we propose a shape, resonant frequency (n=2) and thermoelastic dissipation analysis of free-form microhemispherical shells based on a forming process model. The forming process model is established through process mechanism analysis, which can give the actual shape of the microhemispherical shell under different preforming parameters. Based on this analytical model, the effects of the inner diameter of the annular groove ($r_{i}$), the depth of the annular groove ($h_{c}$), the internal and external pressure ratio ($P_{i}/P_{o}$) and the thickness of the device layer (*h*) on shape, resonant frequency (n=2) and thermoelastic dissipation of microhemispherical shells are studied in detail through fluid–structure coupling analysis and eigenfrequency analysis. The rest of this paper is organized as follows: Section 2 gives the analysis of the blowing mechanism of free-form microhemispherical shells. The process mechanism and process flow of the free-forming method are introduced in detail. Then, the forming process simulation model is established based on a two-dimensional axisymmetric model. The influence of key preforming parameters on the shape of microhemispherical shells is analyzed in Section 3. Section 4 and Section 5 present the analysis of the impact of key preforming parameters on the resonant frequency (n=2) and the thermoelastic dissipation of microhemispherical shells, respectively. Finally, Section 6 concludes this paper with a summary.

## 2. Analysis of the Blowing Mechanism of Free-Form Microhemispherical Shells

As shown in Figure 1, the blowing process is based on the pressure difference between the inside and outside of the cavity after bonding. When heated to above the softening point of fused silica, the fused silica will bulge outward under the action of surface tension until the internal and external pressures are balanced, forming a microhemispherical shell. The pressure difference can be formed by

pressurizing the inside of the cavity, such as using a foaming agent;decompressing the outside of the cavity, such as pumping.

The fabrication process for microhemispheric shells based on the surface-tension method is given in Figure 2 and mainly consists of five steps [20,21].

(a)Masking.(b)Etching annular grooves in the substrate.(c)Bonding the device layer to the substrate layer. For methods using a foaming agent, add the foaming agent prior to this step.(d)Heating and foaming. For methods using external pumping, pumping is performed before this step.(e)Releasing the microhemispherical shell by laser or lapping.

It can be seen that the shape of the microhemispherical shell prepared by the surface tension process is a one-time molding, and the shape is determined by the inner diameter, outer diameter, depth, inner and outer pressure and glass thickness of the annular groove of the substrate before molding.

When the pressure inside and outside the annular groove is balanced, according to Boyle’s Law, we get
(1)$$P_{i}^{{}^{\prime }}V_{c}=P_{o}^{{}^{\prime }}\left(V_{c}\pm V_{s}\right),$$
where $V_{c}$ is the volume of the inner cavity of the annular groove, $V_{s}$ is the inner cavity volume of the microhemispherical shell, and $P_{i}^{{}^{\prime }}$ and $P_{o}^{{}^{\prime }}$ are the inner and outer pressures, respectively, of the annular groove when the softening point is reached.

Since the volume of the gas does not change during the heating process before reaching the softening point, it satisfies Charles’s Law, so the pressure inside and outside the annular groove can be expressed as
(2)$$\left\{\begin{array}{c} P_{i}^{{}^{\prime }}=\left(1+\beta T\right)P_{i},\\ P_{o}^{{}^{\prime }}=\left(1+\beta T\right)P_{o}, \end{array}\right.$$
where, $P_{i}$, $P_{o}$ are the initial pressure inside and outside the annular groove, respectively, β is the thermal expansion coefficient of the gas, and *T* is the temperature change.

From Equations (Equation 1) and (Equation 2), the volume of the inner cavity of the microhemispherical shell is
(3)$$V_{s}=\pi \left(\frac{P_{i}}{P_{o}}-1\right)\left({r_{o}}^{2}-{r_{i}}^{2}\right)h_{c}.$$

Since the bottom surface radius of the microhemispherical shell is determined by the outer diameter of the substrate annular groove, when the outer diameter of the substrate annular groove is fixed, the inner wall volume of the microhemispherical shell is positively related to its shell height, that is, $V_{S}\propto h_{s}$. Therefore,
(4)$$h_{s}\propto \pi \left(\frac{P_{i}}{P_{o}}-1\right)\left({r_{o}}^{2}-{r_{i}}^{2}\right)h_{c}.$$

It can be seen from Equation (Equation 4) that the height of the microhemispherical shell is positively correlated with the depth, outer diameter and pressure ratio of the annular groove, and negatively correlated with the inner diameter of the annular groove. At the same time, changes in the height of the shell will cause changes in the thickness of the shell. In addition, the device layer thickness will also affect the shell thickness.

Since the microhemispherical shell is formed at one time, and the thickness of the shell changes non-uniformly during the molding process, the change of the thickness of the shell at the same time as the height changes needs to be further disclosed.

## 3. Influence of Key Preforming Parameters on the Shape of Microhemispherical Shells

Figure 3 shows the key structural parameters before and after forming. This section will analyze the influence of depth, inner diameter, outer diameter and pressure ratio of the annular groove on the key topographic parameters of the microhemispherical shell, such as height, thickness of top edge ($t_{t}$), thickness of bottom outer edge ($t_{bo}$), radius of bottom support column ($r_{bi}$), thickness of bottom support column ($t_{bi}$), etc., based on the blowing process simulation model.

The blowing process simulation model is established based on the two-dimensional axisymmetric model and is simulated with a Newtonian isothermal fluid flow model. The structural parameters are shown in Figure 3, and the boundary setting is shown in Figure 4, where,

Boundary 1: External pressure of cavity.Boundary 2: Real-time pressure inside the cavity.Boundary 3: Symmetry axis, specify *r* displacement as 0.Boundary 4: Fixed boundaries.

Fused silica has proven to be an excellent material for obtaining high-performance microhemispherical shell resonators [8,9,22]. On the one hand, fused silica is an isotropic material, and the microhemispherical shell can achieve a fully symmetrical structure based on three-dimensional processing, which can achieve better structural symmetry. On the other hand, fused silica has low internal losses, making it easier to achieve high quality factor microhemispherical shells. The material parameters of fused silica are shown in Table 1 [19].

The initial values of the depth, inner diameter, outer diameter, pressure ratio of the annular groove and the thickness of device layer are shown in Table 2.

Table 3, Table 4, Table 5 and Table 6 show the changes of the shape and key shape parameters of the microhemispherical shell with the inner and outer pressure ratio, depth, inner diameter of the annular groove of the substrate and the thickness of the device layer.

From the data in the above tables it can be seen that: The height of the microhemispherical shell is mainly affected by the depth of the substrate annular groove ($h_{c}$) and the internal and external pressure ratio ($P_{i}/P_{o}$), and with an increase of the depth and the pressure ratio, the height of the microhemispherical shell increases. At the same time, as the height increases, the thickness of the top and the edge ($t_{t}$, $t_{bo}$) as well as the radius of the bottom support column ($r_{bi}$) of the shell decreases, while the variation in the thickness of the bottom support column ($t_{bi}$) is small. The inner diameter of the annular groove ($r_{i}$) mainly affects the shape of the support column and also has a slight influence on the height of the microhemispherical shell. Specifically, as the inner diameter of the annular groove ($r_{i}$) increases, the radius of the bottom support column ($r_{bi}$) increases, but the thickness of the bottom support column ($t_{bi}$) changes little, so the hollow radius of the center of the support column becomes larger and larger. At the same time, the height of the microhemispherical shell ($h_{s}$) will be reduced. In addition, it has little effect on the thickness of the top and edge of the microhemispherical shell. The thickness of the device layer (*h*) affects the thickness of the shell, and has less effect on the height of the shell ($h_{s}$) and the radius of the support column ($r_{bi}$). With the increase of the thickness of the device layer, the thickness of the top, the edge and the bottom support column ($t_{t}$, $t_{bo}$ and $t_{bi}$) all increase.

## 4. Simulation Analysis of Resonant Frequency of Microhemispherical Shell

The microhemispherical resonant gyroscope works based on the Coriolis effect and usually operates in n=2 wineglass mode (four antinodes), as shown in Figure 5a. Spatially, the complete vibration of the microhemispherical resonator takes the form of a standing wave. Ideally, the two modal frequencies are exactly the same, and the modal principal axes differ by 45${}^{\circ }$.

When the gyroscope is working, one mode is excited as the driving mode, and the other mode is not excited and is the detection mode. When there is an angular velocity input around the central axis of the microhemispherical shell, under the effect of Coriolis force, the energy of the driving mode will be coupled to the detection mode so that the detection mode is excited, and the resonator behaves as driving mode and detect modal coupling. This causes rotation of the microhemispherical shell vibration to lag behind the rotation of the hemispherical shell, which is called the precession effect, as is shown in Figure 5b [23,24]. Because this effect was first discovered by the British physicist G.H. Byran, it is often said that this effect is the Bryan effect [25]. The proportional relationship between the angle at which the resonator mode shape rotation lags behind the shell rotation (θ) and the shell rotation angle (φ) can be expressed as [26]
(5)θ=−kφ
where *k* is the precession factor of the resonator mode. By measuring the lag angle between the vibration mode and the micro hemispherical shell (θ), the rotation angle of the micro hemispherical shell (φ) can be obtained. Therefore, the micro hemispherical shell also has the special advantage of direct output angle.

The shape of the microhemispherical shell will cause changes in many of its performance parameters, one of which is the effect of the resonant frequency. Because the shape of the microhemispherical shell prepared by the blowing process is of non-equal radius and non-equal thickness, its resonant frequency is difficult to solve by analytical expression. In this section, based on the analysis model of the forming process, the resonant frequencies of microhemispherical shells under different initial parameters are analyzed by means of finite element simulation.

Microhemispherical shell resonators need both small size and sufficient capacitance. At present, the bottom radius of micro hemispherical shell resonators prepared by most blowing processes is generally 3.5 mm to 5 mm [8,11,27,28,29,30]. In this paper, the radius of the bottom surface is taken as 3.5 mm, 4.0 mm and 4.5 mm.

Since the two resonant frequency values for n=2 are exactly the same, only one value is given in this paper. Resonant frequencies (*n* = 2) of microhemispherical shells with different parameters are shown in Figure 6. The results show that the resonant frequency is greatly affected by the depth of the substrate annular groove ($h_{c}$), the internal and external pressure ratio ($P_{i}/P_{o}$) and the thickness of the device layer (*h*), while it is less affected by the inner diameter of the annular groove ($r_{i}$), which is basically consistent with the influence of $h_{c}$, $P_{i}/P_{o}$, *h* and $r_{i}$ on the shape of the microhemispherical shell. Moreover, for different outer diameters of annular grooves (that is, the radius of the bottom surface of the microhemispherical shell), the influence rules are the same.

Specifically, as the thickness of the fused silica layer (*h*) increases, the resonant frequency of the microhemispherical shell (n=2) increases, which is caused by the increase of the thickness of the microhemispherical shell when the thickness of the fused silica layer increases. At the same time, the increase in the resonant frequency is approximately linear with the increase in the thickness of the fused silica layer. When *h* changes from 0.2 mm to 0.4 mm, the resonant frequency increases from 9.1 kHz to 17.8 kHz (r=3.5 mm), from 7.8 kHz to 15.2 kHz (r=4 mm) and from 6.8 kHz to 13.2 kHz (r=4.5 mm). On the contrary, an increase in the depth of the substrate annular groove ($h_{c}$) or of the internal and external pressure ratio ($P_{i}/P_{o}$) will decrease the resonant frequency, which is caused by the increase of the height and the decrease of the thickness of the microhemispherical shell when $h_{c}$ or $P_{i}/P_{o}$ increase. When $h_{c}$ increases from 0.2 mm to 0.5 mm for r=3.5 mm, 4 mm and 4.5 mm, the resonant frequency decreases by 9.2 kHz, 10.4 kHz and 7.4 kHz, respectively; similarly, when $P_{i}/P_{o}$ increases from 6 to 13 for r=3.5 mm, 4 mm and 4.5 mm, the resonant frequency is reduced by 11.4 kHz, 9.9 kHz and 6.2 kHz, respectively.

## 5. Simulation Analysis of Thermoelastic Dissipation of Microhemispherical Shell

The 3D thermoelastic damping analysis model for isotropic materials is given in Equation (Equation 6), which shows the thermoelastic behavior of an isotropic solid material [15,31]. The model describes the coupling between the temperature field and the displacement field, and the coupling between the strain field and the temperature field, which reflects the influence of the temperature field on the displacement field of the harmonic oscillator and the strain field generated when the structure vibrates as a result of the effects of temperature gradients.
(6)$$\left\{\begin{array}{cc} \rho \frac{\partial^{2}u}{\partial t^{2}}= & \frac{E}{2(1+v)}\left(\frac{\partial^{2}u}{\partial x^{2}}+\frac{\partial^{2}u}{\partial y^{2}}+\frac{\partial^{2}u}{\partial z^{2}}\right)+\frac{E}{2(1+v)(1-2v)}\left(\frac{\partial^{2}u}{\partial x^{2}}+\frac{\partial^{2}v}{\partial x\partial y}+\frac{\partial^{2}w}{\partial x\partial z}\right)-\frac{E\alpha }{1-2v}\frac{\partial T}{\partial x}\\ \rho \frac{\partial^{2}v}{\partial t^{2}}= & \frac{E}{2(1+v)}\left(\frac{\partial^{2}v}{\partial x^{2}}+\frac{\partial^{2}v}{\partial y^{2}}+\frac{\partial^{2}v}{\partial z^{2}}\right)+\frac{E}{2(1+v)(1-2v)}\left(\frac{\partial^{2}u}{\partial y\partial x}+\frac{\partial^{2}v}{\partial y^{2}}+\frac{\partial^{2}w}{\partial y\partial z}\right)-\frac{E\alpha }{1-2v}\frac{\partial T}{\partial y}\\ \rho \frac{\partial^{2}w}{\partial t^{2}}= & \frac{E}{2(1+v)}\left(\frac{\partial^{2}w}{\partial x^{2}}+\frac{\partial^{2}w}{\partial y^{2}}+\frac{\partial^{2}w}{\partial z^{2}}\right)+\frac{E}{2(1+v)(1-2v)}\left(\frac{\partial^{2}u}{\partial z\partial x}+\frac{\partial^{2}v}{\partial z\partial y}+\frac{\partial^{2}w}{\partial z^{2}}\right)-\frac{E\alpha }{1-2v}\frac{\partial T}{\partial z}\\ k\nabla^{2}T= & \rho C_{SP}\frac{\partial T}{\partial t}+\frac{E\alpha T_{0}}{1-2v}\left(\frac{\partial \dot{u}}{\partial x}+\frac{\partial \dot{v}}{\partial y}+\frac{\partial \dot{w}}{\partial z}\right) \end{array}\right.$$
where ρ, *E*, μ, α, $C_{SP}$ and *k* are density, Young’s modulus, Poisson’s ratio, coefficient, specific heat capacity and thermal conductivity of the material, respectively, *u*, *v* and *w* are displacements in the *x*, *y* and *z* directions, respectively.

For a 3D structure, the quality factor is difficult to calculate analytically and can be estimated using
(7)$$Q_{i}=\frac{Re(\omega_{i})}{2Im(\omega_{i})},$$
where, $Q_{i}$ and $\omega_{i}$ are the quality factor and eigenfrequency, respectively, of the *i*th mode [12]. Re and Im represent the real and imaginary parts of the eigenfrequency, respectively.

In this section, fused silica is chosen as the device material. The finite element simulation analysis is based on thermoelasticity physics. At the same time, the *Q* of the microhemispherical shell resonator is simulated assuming TED is the only dominant dissipation mechanism. Figure 7 gives the $Q_{TED}$ of microhemispherical shells with different parameters. $Q_{TED}$ is the *Q* due to TED.

The results illustrate that:
When $r_{i}$ increases from 0.5 mm to 1.0 mm for r=3.5 mm, 4 mm and 4.5 mm, the $Q_{TED}$ decreases by 108.8 million, 113.6 million and 128.7 million, respectively;When *h* increases from 0.2 mm to 0.4 mm for r=3.5 mm, 4 mm and 4.5 mm, the $Q_{TED}$ increases by 616.2 million, 619.1 million and 636.6 million, respectively;When $h_{c}$ increases from 0.2 mm to 0.5 mm for r=3.5 mm, 4 mm and 4.5 mm, the $Q_{TED}$ decreases by 385.3 million, 330.0 million and 325.9 million, respectively;When $P_{i}/P_{o}$ increases from 7 to 13 for r=3.5 mm, 4 mm and 4.5 mm, the $Q_{TED}$ decreases by 386.1 million, 336.6 million and 254.5 million, respectively.

The results demonstrate that the inner diameter of the annular groove ($r_{i}$), the depth of the annular groove ($h_{c}$), the internal and external pressure ratio ($P_{i}/P_{o}$) and the thickness of the device layer (*h*) all affect the thermoelastic dissipation of the microhemispherical shell. Specifically, the $Q_{TED}$ decreases with the increase of $r_{i}$, $h_{c}$ and $P_{i}/P_{o}$, and increases with the increase of *h*. The reasons are that the increase of $r_{i}$ leads to the increase of the radius of the support column and the increase of the hollowness, resulting in the decrease of $Q_{TED}$. The increase of *h* increases the thickness of the microhemispherical shell, resulting in the increase of $Q_{TED}$. The increase of $h_{c}$ and $P_{i}/P_{o}$ increase the height and decrease the thickness of the microhemispherical shell, resulting in decreased $Q_{TED}$.

## 6. Conclusions

The simulation and modeling results presented above have provided the impact of the inner diameter of the annular groove ($r_{i}$), the depth of the annular groove ($h_{c}$), the internal and external pressure ratio ($P_{i}/P_{o}$) and the thickness of the device layer (*h*) on shape, resonant frequency (n=2) and thermoelastic dissipation of microhemispherical shells.

In this paper, a process analysis model is established by analyzing the blowing process of the free-form method. Based on this process model, a more accurate microhemisphere topography is obtained compared to the equivalent method. The influence of design parameters before forming the microhemispherical shell shape was studied in detail. Based on microhemispherical shell shape under different design parameters, modal analysis and thermal stress-based eigenfrequency analysis were carried out to explain the influence of design parameters on resonance frequency and thermoelastic dissipation.

The analysis results of the microhemispherical shell shape show that $h_{c}$ and $P_{i}/P_{o}$ affect the height and thickness of the microhemispherical shell, *h* affects the thickness of the microhemispherical shell, and $r_{i}$ mainly affects the radius of the support column and also has a slight effect on the height of the microhemispherical shell. In terms of the influence on the resonant frequency (n=2), the increase of $h_{c}$ and $P_{i}/P_{o}$ increase the height and decrease the thickness of the microhemispherical shell, which decreases the resonance frequency; the increase of *h* increases the thickness of the microhemispherical shell, which increases the resonant frequency; the increase of $r_{i}$ has less influence on the height and thickness of the microhemispherical shell and thus has less influence on the resonant frequency. In terms of thermoelastic dissipation, $h_{c}$ and $P_{i}/P_{o}$ decrease $Q_{TED}$, *h* increases $Q_{TED}$, and $r_{i}$ also decreases $Q_{TED}$, but the magnitude is smaller than that of $h_{c}$ and $P_{i}/P_{o}$.

The research in this paper can provide theoretical guidance for the comprehensive design of the parameters of free-form microhemispheric shells before forming. Since the fused silica material itself does not have electrical conductivity, when the microhemispherical shell is applied to the microhemispheric resonant gyroscope, surface metallization is required, which will lead to changes in the thermoelastic dissipation. The impact analysis will be future work.

## Figures and Tables

**Figure 1 micromachines-13-00913-f001:**
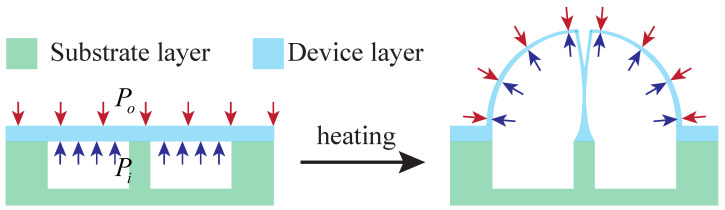
Cross-sectional schematic diagram of the blowing process.

**Figure 2 micromachines-13-00913-f002:**
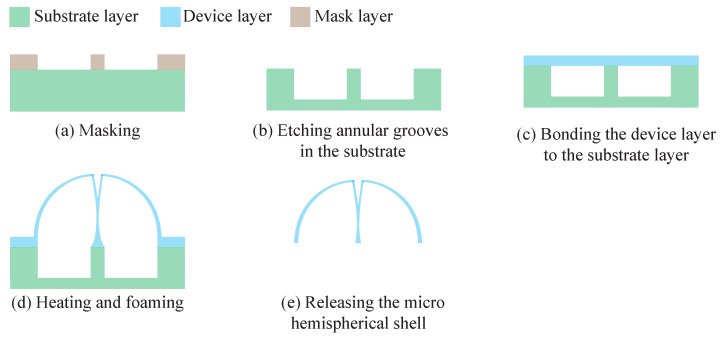
Concept schematic of fabrication process for microhemispheric shell based on the surface tension method.

**Figure 3 micromachines-13-00913-f003:**
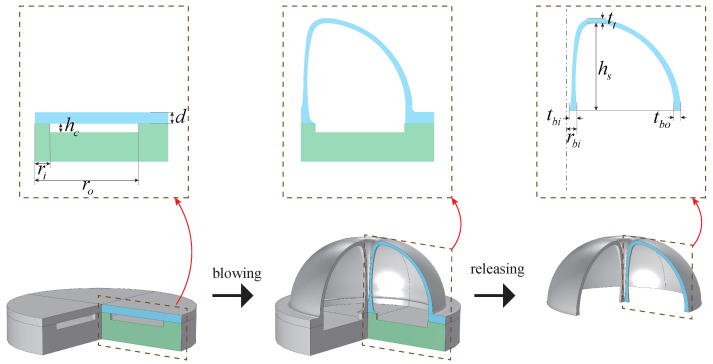
Microhemispherical shell-forming process and key structural parameters before and after forming.

**Figure 4 micromachines-13-00913-f004:**
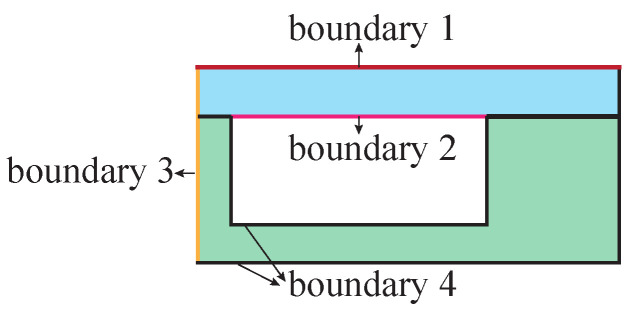
Microhemispherical shell forming process and key structural parameters before and after forming.

**Figure 5 micromachines-13-00913-f005:**
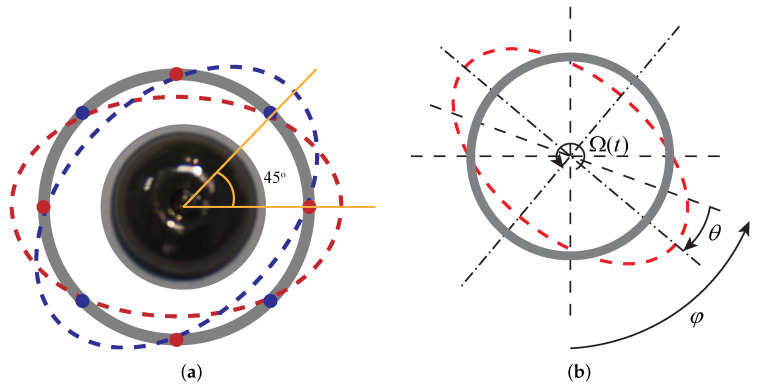
The modes and precession principle of the microhemispherical shell resonator (n=2). (**a**) The n=2 modes of the microhemispherical shell resonator. (**b**) The precession principle of hemispherical resonant gyroscope.

**Figure 6 micromachines-13-00913-f006:**
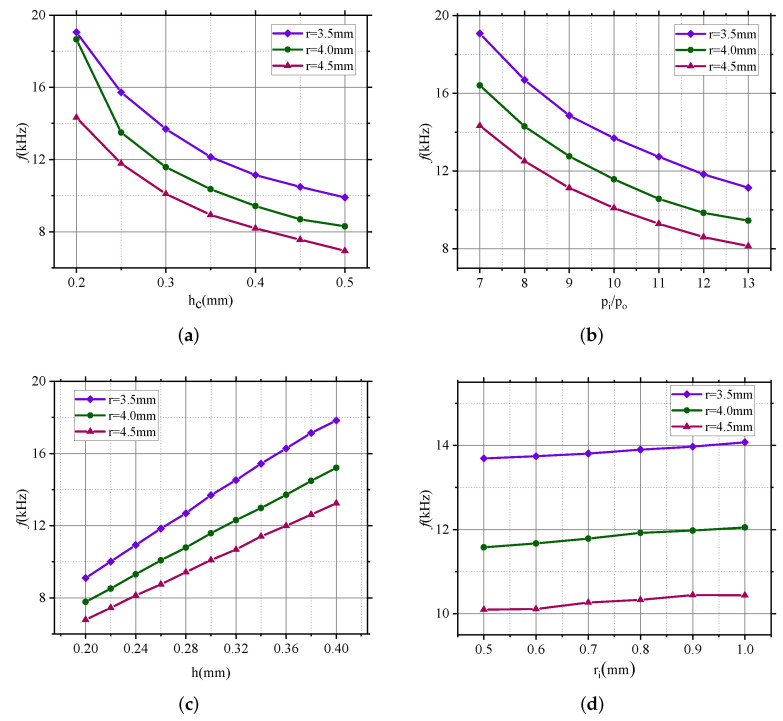
Resonant frequencies (n=2) of microhemispherical shells with different parameters. (**a**) Influence of groove depth on resonant frequency. (**b**) Influence of pressure ratio inside and outside the groove on resonant frequency. (**c**) Influence of fused silica thickness on resonant frequency. (**d**) Influence of groove inner diameter on resonance frequency.

**Figure 7 micromachines-13-00913-f007:**
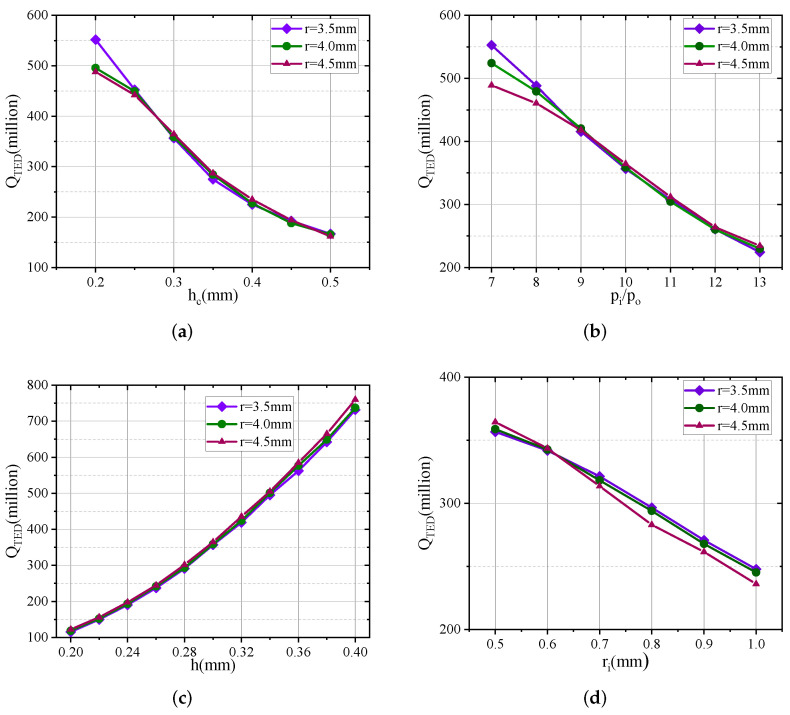
$Q_{TED}$ of microhemispherical shells with different parameters. (**a**) Influence of groove depth on $Q_{TED}$. (**b**) Influence of pressure ratio inside and outside the groove on $Q_{TED}$. (**c**) Influence of fused silica thickness on $Q_{TED}$. (**d**) Influence of groove inner diameter on $Q_{TED}$.

**Table 1 micromachines-13-00913-t001:** Material parameters of fused silica.

Parameter	Value	Unit
Young’s modulus	70	GPa
Poisson’s ratio	0.17	–
Density	2200	kg/m^3^
Thermal conductivity	1.4	W/(m·K)
Specific heat capacity	730	J/(kg × K)
Coefficient of thermal expansion	$5\times 10^{-7}$	1/K

**Table 2 micromachines-13-00913-t002:** Initial values of the depth, inner diameter, outer diameter, pressure ratio of the annular groove and the thickness of device layer.

Parameter	Value	Unit
depth of the annular groove ($h_{c}$)	0.3	mm
inner diameter of the annular groove ($r_{i}$)	0.5	mm
outer diameter of the annular groove ($r_{o}$)	3.5	mm
pressure ratio of the annular groove ($P_{i}/P_{o}$)	10	–
thickness of device layer (*h*)	0.3	mm

**Table 3 micromachines-13-00913-t003:** Shape and key shape parameters of microhemispherical shells under different pressure ratios.

$P_{i}/P_{o}$	7	9	11	13
Shape	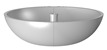	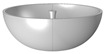	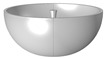	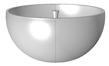
$h_{s}$ (mm)	2.002	2.711	3.281	3.787
$r_{bi}$ (μm)	320.4	288.4	282.2	273.1
$t_{bi}\;\left(\mu m\right)$	169.5	176.8	182.3	180.8
$t_{bo}\;\left(\mu m\right)$	211.8	192.9	185.4	175.9
$t_{t}\;\left(\mu m\right)$	160.9	133.6	119.0	105.8

**Table 4 micromachines-13-00913-t004:** Shape and key shape parameters of microhemispherical shells at different groove depths.

$h_{c}$ (mm)	0.2	0.3	0.4	0.5
Shape	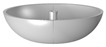	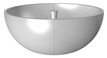	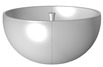	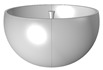
$h_{s}$ (mm)	2.003	3.007	3.788	4.368
$r_{bi}\;\left(\mu m\right)$	321.9	285.6	274.0	271.5
$t_{bi}\;\left(\mu m\right)$	167.9	181.9	181.3	181.5
$t_{bo}\;\left(\mu m\right)$	211.0	189.7	175.8	172.0
$t_{t}\;\left(\mu m\right)$	161.9	127.4	104.5	96.2

**Table 5 micromachines-13-00913-t005:** Shape and key shape parameters of microhemispherical shells at different groove inner diameters.

$r_{i}$ (mm)	0.5	0.6	0.7	0.8
Shape	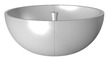	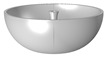	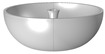	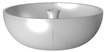
$h_{s}$ (mm)	3.007	2.869	2.738	2.618
$r_{bi}\;\left(\mu m\right)$	285.6	375.8	471.5	564.6
$t_{bi}\;\left(\mu m\right)$	181.9	177.3	172.1	172.2
$t_{bo}\;\left(\mu m\right)$	189.7	187.1	184.4	185.0
$t_{t}\;\left(\mu m\right)$	127.4	124.5	122.6	122.2

**Table 6 micromachines-13-00913-t006:** Shape and key shape parameters of microhemispherical shells at different thicknesses of device layer.

*h* (mm)	0.22	0.26	0.30	0.34
Shape	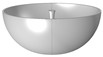	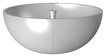	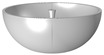	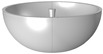
$h_{s}$(mm)	3.056	3.029	3.007	3.006
$r_{bi}\;\left(\mu m\right)$	270.2	283.1	285.6	272.5
$t_{bi}\;\left(\mu m\right)$	93.3	131.3	181.9	246.1
$t_{bo}\;\left(\mu m\right)$	127.5	155.9	189.7	229.3
$t_{t}\;\left(\mu m\right)$	91.1	108.6	127.4	143.9

## Data Availability

Not applicable.

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
