# Peer review of "Shape, Resonant Frequency and Thermoelastic Dissipation Analysis of Free-Formed Microhemispherical Shells Based on Forming Process Modeling"

_micromachines, 2022, doi:10.3390/mi13060913_

Round 1
Reviewer 1 Report
The paper would be of much higher interest, if the simulations would be compared with experiments as a validation of the simulation results. The presented fabrication process is very special, therefore the results of the simulations are not of general interest but only for this very special technology together with the chosen design of the microhemispherical shell.
In general the chosen methods are correct, but if the details of the models and the simulations are correct can not be evaluated. Therefore a comparison with experiments and measurement results would be of great interest. For example the resonant frequency of two or three different geometries. Moreover, the variation of the radius of the shell with only two values (3.5 and 4.0 mm) is very low. No scaling systematic can be deduced from that. Therefore the significance of the content for other scientists is at most average. An experimental validation would enhance the soundness and significance of the paper very much. If it is not possible to fabricate the structures to measure the main parameters of the resonator, than at least a broader variation of the geometric parameters would be desirable, so that at least the design rules can be
Author Response
Thank you for your thorough review and insightful comments.
We addressed all your comments and modified the manuscript as described below. The corresponding changes were highlighted in red in the manuscript. Below, the original comments are in black, and our responses are in red.
The paper would be of much higher interest, if the simulations would be compared with experiments as a validation of the simulation results. The presented fabrication process is very special, therefore the results of the simulations are not of general interest but only for this very special technology together with the chosen design of the micro-hemispherical shell.
In general the chosen methods are correct, but if the details of the models and the simulations are correct can not be evaluated. Therefore a comparison with experiments and measurement results would be of great interest. For example the resonant frequency of two or three different geometries. Moreover, the variation of the radius of the shell with only two values (3.5 and 4.0 mm) is very low. No scaling systematic can be deduced from that. Therefore the significance of the content for other scientists is at most average. An experimental validation would enhance the soundness and significance of the paper very much. If it is not possible to fabricate the structures to measure the main parameters of the resonator, than at least a broader variation of the geometric parameters would be desirable, so that at least the design rules can be
- Added the description of the simulation method:
(1)Line 146: The blowing process simulation model a Newtonian isothermal fluid flow model using COMSOL Multiphysics FE package.
(2) Line 238: The finite element simulation analysis for TED is based on the thermoelasticity physics.
- Experimental comparison
Experimental comparisons are very important. However, the material of the micro-hemispherical shell is fused silica, which does not have electrostatic properties. Therefore, the micro-hemispherical shell requires surface metallization before testing and characterization. The surface metallization will affect the resonant frequency and thermal elastic dissipation. In addition, encapsulation stress and vacuum will also affect the micro-hemispherical shell. Currently, we are conducting research in the surface metallization. Therefore, based on the molding model, the influence mechanism of each link needs to be fully studied, and finally a systematic experimental comparison will be carried out.
- The radius size of the micro-hemispherical shell
(1) Sections 4 and 5: Added a set of simulation data (r=4.5mm).
(2) Basis for the value range of r.
In 2010, the Defense Advanced Research Projects Agency (DARPA) launched the DARPABAA-10-39 Microscale Rate Integrating Gyroscope program, and dozens of organizations began to develop miniaturized hemispherical resonant gyroscopes. The volume is not more than 1 cubic centimeter. In addition, the size of the micro-hemispherical shell will affect its structural rigidity on the one hand, and the capacitance formed between it and the substrate on the other hand. If the size is too small, the structural stiffness will be too large and the excitation capacitance will be too small, causing vibration excitation and detection challenges.
At present, the micro-hemispherical shell resonators of some representative institutions are 5mm (University of Michigan, the micro-hemispheric resonant gyroscope currently developed has the best performance)[1-3], 3.5mm (University of California, Irvine)[4], 3.8mm and 3.95mm (National University of Defense Technology)[5,6], 3.5mm (Southeast University)[7].
[1] Nagourney T, Cho J Y, Shiari B, et al. 259 Second ring-down time and 4.45 million quality factor in 5.5 kHz fused silica birdbath shell resonator[C]. In: International Conference on Solid-State Sensors, Actuators and Microsystems. 2017. 790–793.
[2] Singh S, Nagourney T, Cho J Y, et al. Design and fabrication of high-Q birdbath resonator for MEMS gyroscopes[C]. In: Position, Location and Navigation Symposium (PLANS), 2018 IEEE/ION. 2018. 15–19.
[3] Cho J Y, Singh S, Woo J, et al. 0.00016 deg/√hr Angle Random Walk (ARW) and 0.0014 deg/hr Bias Instability (BI) from a 5.2M-Q and 1-cm Precision Shell Integrating (PSI) Gyroscope[C] // 2020 IEEE International Symposium on Inertial Sensors and Systems (INERTIAL). 2020 : 1 – 4.
[4] Senkal, D., Ahamed, M. J., Ardakani, M. H. A., Askari, S., and Shkel, A. M. Demonstration of 1 Million-Factor on Microglassblown Wineglass Resonators With Out-of-Plane Electrostatic Transduction[J]. IEEE J. Microelectromech. Syst., 2015, 24(1): 29–37.
[5] Xiao D , Wei L , Hou Z , et al. Fused Silica Micro Shell Resonator With T-Shape Masses for Gyroscopic Application[J]. Journal of Microelectromechanical Systems, 2017, PP(99):1-12.
[6] Y. Shi, K. Lu, X. Xi, Y. Wu, D. Xiao, and X. Wu, "Geometric Imperfection Characterization and Precise Assembly of Micro Shell Resonators," J. Microelectromech. Syst., vol. 29, pp. 480-489, 2020.
[7] Luo B, Shang J, ZhangY. Hemispherical glass shell resonators fabricated using Chemical Foaming Process[C]. In: Electronic Components and Technology Conference (ECTC), 2015 IEEE 65th. 2015. 2217–2221.
Furthermore, the radius size is easily determined initially based on the overall size to be designed. However, the determination of hc, pi/po, h, and ri needs to be comprehensively determined around the expected topographic features, resonant frequency, and thermoelastic loss. The analysis in this paper provides the basis for the determination of these parameters.

Reviewer 2 Report
The authors investigated the free-form micro-hemispherical shell's frequency and thermoelastic dissipation analysis. The concept and work are applicable and deserve to be considered for the further process. However, some major comments should be answered before the final review decision. Some of the critical questions are listed below:
1- I did not understand if the work was experimental or simulated? If you made a setup for this issue (experimental), why didn't you provide some real pictures for this? Alternatively, if the work is only a simulation, please present more information regarding the methodology. I am aware of mentioning it in the second paragraph in the conclusion part as "theoretical -line 250"; however, the theoretical shreds of evidence are weak as the deriving governing equations, solving method, etc.
2- You studied the gyroscopic effects of the analysis. Only the study is referred to the Bryan effect [16] in line 162. It is a little confusing for readers. Please present some more information about this issue, and just more explanations can be referenced as [16].
3- Equation (5) has been referred to [17 and 18]. If a schematic view is set into the manuscript, it can be so helpful in understanding the formula and what are displacements u, v, and w.
4- Is the thermal conductivity coefficient α constant? I think it is constant in an interval. However, it should be varied beyond the interval.
5- Your title contains “thermoelastic analysis." However, I did not see any thermoelastic analysis in the figures in which the temperature effect has been investigated.
6- Why did you consider fused silica as your primary material? What are the benefits of choosing this material?
7- The introduction part can list other references for supporting the work. Some of them are expressed below:
· https://ieeexplore.ieee.org/document/6994789
· https://www.sciencedirect.com/science/article/abs/pii/S0020722520300240
· https://ieeexplore.ieee.org/document/7935681
· https://www.sciencedirect.com/science/article/abs/pii/S0304399117302279
Please provide clear and suitable answers to the mentioned comments.
Author Response
Response to Reviewers
Reviewer #2
Thank you for your thorough review and insightful comments.
We addressed all your comments and modified the manuscript as described below. The corresponding changes were highlighted in red in the manuscript. Below, the original comments are in black, and our responses are in red.
The authors investigated the free-form micro-hemispherical shell's frequency and thermoelastic dissipation analysis. The concept and work are applicable and deserve to be considered for the further process. However, some major comments should be answered before the final review decision. Some of the critical questions are listed below:
1- I did not understand if the work was experimental or simulated? If you made a setup for this issue (experimental), why didn't you provide some real pictures for this? Alternatively, if the work is only a simulation, please present more information regarding the methodology. I am aware of mentioning it in the second paragraph in the conclusion part as "theoretical -line 250"; however, the theoretical shreds of evidence are weak as the deriving governing equations, solving method, etc.
(1) Introduction: Added references around TED research of resonators. Most of the references have adopted the method of finite element simulation.
This work was simulated by finite element analysis. The blowing process modal was established and analyzed by using the Newtonian isothermal fluid flow model of COMSOL Multiphysics FE package. And the resonant frequency and thermoelastic analysis were carried out by using eigenfrequency and thermoelastic components of COMSOL Multiphysics FE package. The methods used are mature and widely used in MEMS device research.
(2) Sections 4 and 5: Added a set of simulation data (r=4.5mm).
2- You studied the gyroscopic effects of the analysis. Only the study is referred to the Bryan effect [16] in line 162. It is a little confusing for readers. Please present some more information about this issue, and just more explanations can be referenced as [16].
Sections 4: Added a detailed description of the working principle of micro-hemispherical resonant gyroscope.
3- Equation (5) has been referred to [17 and 18]. If a schematic view is set into the manuscript, it can be so helpful in understanding the formula and what are displacements u, v, and w.
u, v, and w is the displacement of any differential unit in the three-dimensional resonator in the x, y, and z directions. x, y, and z are Cartesian coordinate systems, which can be defined independently.
4- Is the thermal conductivity coefficient α constant? I think it is constant in an interval. However, it should be varied beyond the interval.
Generally, the thermal conductivity of a material varies with temperature. In this paper, the blowing simulation topography is mainly affected by viscosity and density. In the resonance frequency analysis, the resonance frequency is mainly affected by the elastic modulus and density. Thermal conductivity needs to be set only in thermoelastic loss simulations. On the one hand, the thermal conductivity of fused silica is extremely low, and on the other hand, the thermal conductivity changes very little over a small range of temperature changes. Therefore, the parameters at normal temperature are used in the thermoelastic simulation, that is, the thermal conductivity is set as a constant.
5- Your title contains “thermoelastic analysis." However, I did not see any thermoelastic analysis in the figures in which the temperature effect has been investigated.
Thermoelastic dissipation (TED) is characterized by causing a change in the quality factor of the resonator. In research, QTED is often used to represent the level of thermoelastic dissipation. The larger the QTED, the smaller the thermoelastic dissipation; conversely, the smaller the QTED, the larger the thermoelastic dissipation.
6- Why did you consider fused silica as your primary material? What are the benefits of choosing this material?
Lines 153-158: Added instructions for selecting fused silica material.
On the one hand, fused silica is an isotropic material, and the micro-hemispherical shell can achieve a fully symmetrical structure based on a three-dimensional processing technology, which can achieve better structural symmetry. On the other hand, fused silica has low internal losses, making it easier to achieve high quality factor micro-hemispherical shells. The macro-sized hemispherical resonator material is fused silica, which has an extremely high quality factor. In addition, by investigating the development of micro-hemispherical resonators, it can be seen that the materials for the blowing process are mainly divided into fused silica and glass. The University of Michigan and the National University of Defense Technology used fused silica to fabricate extremely high-performance micro-hemispherical resonators; the University of California, Irvine used glass materials in the early days, and the performance of the resonators was not high. They have also switched to using fused silica as the micro-hemispherical shell material in recent years, and the performance of the resonator has been greatly improved.
7- The introduction part can list other references for supporting the work. Some of them are expressed below:
- https://ieeexplore.ieee.org/document/6994789
- https://www.sciencedirect.com/science/article/abs/pii/S0020722520300240
- https://ieeexplore.ieee.org/document/7935681
- https://www.sciencedirect.com/science/article/abs/pii/S0304399117302279
Added more references supporting this work, including https://ieeexplore.ieee.org/document/6994789, https://www.sciencedirect.com/science/article/abs/pii/S0020722520300240, and https://ieeexplore.ieee.org/document/7935681.
Since the reference in https://www.sciencedirect.com/science/article/abs/pii/S0304399117302279 have little relevance to this article, we are sorry that we can not find a suitable place to quote it.
The added references are as follows
- Dastjerdi, S.; Akgöz, B.; Civalek, Ö. On the Effect of Viscoelasticity on Behavior of Gyroscopes. International Journal of Engineering Science 2020, 149, 103236.
- Nayfeh, A.; Younis, M. Modeling and Simulations of Thermoelastic Damping in Microplates. Journal of Micromechanics and Microengineering 2004, 14, 1711.
- Candler, R.; Duwel, A.; Varghese, M.; Chandorkar, S.; Hopcroft, M.; Park, W.T.; Kim, B.; Yama, G.; Partridge, A.; Lutz, M.; et al. Impact of Geometry on Thermoelastic Dissipation in Micromechanical Resonant Beams. Journal of Microelectromechanical Systems 2006, 15, 927–934.
- Duwel, A.; Candler, R.N.; Kenny, T.W.; Varghese, M. Engineering MEMS Resonators With Low Thermoelastic Damping. Journal of Microelectromechanical Systems 2006, 15, 1437–1445.
- Ghaffari, S.; Ng, E.J.; Ahn, C.H.; Yang, Y.; Wang, S.; Hong, V.A.; Kenny, T.W. Accurate Modeling of Quality Factor Behavior of Complex Silicon MEMS Resonators. Journal of Microelectromechanical Systems 2015, 24, 276–288.
- Hossain, S.T.; McWilliam, S.; Popov, A.A. An Investigation on Thermoelastic Damping of High-Q Ring Resonators. International Journal of Mechanical Sciences 2016, 106, 209–219.
- Shiari, B.; Nagourney, T.; Darvishian, A.; Cho, J.Y.; Najafi, K. Numerical Study of Impact of Surface Roughness on Thermoelastic Loss of Micro-Resonators. In Proceedings of the 2017 IEEE International Symposium on Inertial Sensors and Systems (INERTIAL), 2017, pp. 74–77.
- Asadian, M.; Wang, Y.; Shkel, A. Design and Fabrication of 3D Fused Quartz Shell Resonators for Broad Range of Frequencies and Increased Decay Time. 2018.
- Ermakov, R.; Skripal, E.; Kondratov, D.; L’Vov, A.; Seranova, A.; Gutsevich, D. Development of a Vibrational Error Model of a Hemispherical Resonator Gyroscope. 2018, pp. 1–3.
- Zhuravlev, V. Temperature Drift of a Hemispherical Resonator Gyro (HRG). Mechanics of So
- Zhuravlev, V.; Klimov, D.M. The Solid State Wave Gyroscope (In Russian), 1985.
- Xiao, D.; Li, W.; Hou, Z.; Lu, K.; Shi, Y.; Wu, Y.; Wu, X. Fused Silica Micro Shell Resonator With T-Shape Masses for Gyroscopic Application. Journal of Microelectromechanical Systems 2017, PP, 1–12.
- Singh, S.; Nagourney, T.; Cho, J.; Darvishian, A.; Najafi, K.; Shiari, B. Design and Fabrication of High-Q Birdbath Resonator for MEMS Gyroscopes. 2018, pp. 15–19.
- Senkal, D.; Ahamed, M.; Asadian, M.; Askari, S.; Shkel, A. Demonstration of 1 Million Q -Factor on Microglassblown Wineglass Resonators With Out-of-Plane Electrostatic Transduction. Journal of Microelectromechanical Systems 2015, 24, 29–37.
- Nagourney, T.; Cho, J.; Shiari, B.; Darvishian, A.; Najafi, K. 259 Second Ring-down Time and 4.45 Million Quality Factor in 5.5 kHz Fused Silica Birdbath Shell Resonator. 2017, pp. 790–793. doi:10.1109/TRANSDUCERS.2017.7994167.

Reviewer 3 Report
In the article, the authors considered issues related to the possibility of manufacturing a gyroscope with a micro-hemispherical resonator. The resonator was fabricated by blowing free-form shells. The proposed technology will make it possible to reduce the cost of the resonator production process. Therefore, the article is relevant. In this work, the authors studied the effect of the inner diameter and depth of the annular groove, as well as the ratio of internal and external pressure on the resonant frequency of the shell. In addition, the influence of the height and thickness of the microshell on the resonant frequency was considered in the article. Research in this area represent great interest. The disadvantages of the study include the lack of references to the works of authors from Russia, for example, to the works of V.F. Zhuravlev and D. M. Klimov.Author Response
Response to Reviewers
Reviewer #3
Thank you for your thorough review and insightful comments.
We addressed all your comments and modified the manuscript as described below. The corresponding changes were highlighted in red in the manuscript. Below, the original comments are in black, and our responses are in red.
In the article, the authors considered issues related to the possibility of manufacturing a gyroscope with a micro-hemispherical resonator. The resonator was fabricated by blowing free-form shells. The proposed technology will make it possible to reduce the cost of the resonator production process. Therefore, the article is relevant. In this work, the authors studied the effect of the inner diameter and depth of the annular groove, as well as the ratio of internal and external pressure on the resonant frequency of the shell. In addition, the influence of the height and thickness of the microshell on the resonant frequency was considered in the article. Research in this area represent great interest. The disadvantages of the study include the lack of references to the works of authors from Russia, for example, to the works of V.F. Zhuravlev and D. M. Klimov.
Added more references supporting this work, including references to the works of authors from Russia.
- Ermakov, R.; Skripal, E.; Kondratov, D.; L’Vov, A.; Seranova, A.; Gutsevich, D. Development of a Vibrational Error Model of a Hemispherical Resonator Gyroscope. 2018, pp. 1–3.
- Zhuravlev, V. Temperature Drift of a Hemispherical Resonator Gyro (HRG). Mechanics of Solids 2018, 53, 241–248.
- Zhuravlev, V.; Klimov, D.M. The Solid State Wave Gyroscope (In Russian), 1985.

Round 2
Reviewer 2 Report
Authors have responded to the concerns, and the publication of the manuscript is recommended now.